# Pathophysiological Impact of the MEK5/ERK5 Pathway in Oxidative Stress

**DOI:** 10.3390/cells12081154

**Published:** 2023-04-13

**Authors:** Ignazia Tusa, Alessio Menconi, Alessandro Tubita, Elisabetta Rovida

**Affiliations:** Department of Experimental and Clinical Biomedical Sciences “Mario Serio”, University of Florence, 50134 Florence, Italy

**Keywords:** ERK5, KLF2/4, MEF2, oxidative stress, oxidative damage, ROS, RNS, antioxidant response

## Abstract

Oxidative stress regulates many physiological and pathological processes. Indeed, a low increase in the basal level of reactive oxygen species (ROS) is essential for various cellular functions, including signal transduction, gene expression, cell survival or death, as well as antioxidant capacity. However, if the amount of generated ROS overcomes the antioxidant capacity, excessive ROS results in cellular dysfunctions as a consequence of damage to cellular components, including DNA, lipids and proteins, and may eventually lead to cell death or carcinogenesis. Both in vitro and in vivo investigations have shown that activation of the mitogen-activated protein kinase kinase 5/extracellular signal-regulated kinase 5 (MEK5/ERK5) pathway is frequently involved in oxidative stress-elicited effects. In particular, accumulating evidence identified a prominent role of this pathway in the anti-oxidative response. In this respect, activation of krüppel-like factor 2/4 and nuclear factor erythroid 2-related factor 2 emerged among the most frequent events in ERK5-mediated response to oxidative stress. This review summarizes what is known about the role of the MEK5/ERK5 pathway in the response to oxidative stress in pathophysiological contexts within the cardiovascular, respiratory, lymphohematopoietic, urinary and central nervous systems. The possible beneficial or detrimental effects exerted by the MEK5/ERK5 pathway in the above systems are also discussed.

## 1. Introduction

The extracellular signal-regulated kinase 5 (ERK5), also referred to as big mitogen-activated kinase 1 (BMK1), is the last discovered member of the canonical mitogen-activated protein kinase (MAPK) [1,2,3]. As is the case with other MAPKs, a kinase cascade leads to ERK5 activation via a specific upstream dual-specificity kinase, MEK5. The MEK5/ERK5 pathway is involved in survival, antiapoptotic signalling, proliferation and differentiation of many cell types [3] and plays a role in the onset and progression of cancer [4].

Several lines of evidence point to an important role for ERK5 during oxidative stress in cells. In this review, we summarize: i. The molecular mechanisms through which oxidative stress influences MEK5/ERK5 activation; ii. The known downstream targets involved in biological outcomes following oxidative stress-dependent effects on MEK5/ERK5 activity; iii. Whether the MEK5/ERK5 pathway plays a protective or detrimental role in both physiological and pathological processes when cells are undergoing oxidative stress. The latter point is central, since targeting MEK5/ERK5 signalling, alone or in combination with chemotherapeutics or targeted drugs, is increasingly being considered as a possible treatment for cancer and inflammatory diseases [5].

## 2. Oxidative Stress

Oxidative stress occurs when the redox state is altered due to an imbalance between the produced reactive oxygen species (ROS) or reactive nitrogen species (RNS) and the endogenous antioxidant defence mechanisms [6,7,8]. ROS production relies on both enzymatic and non-enzymatic reactions. The superoxide radical (O_2_^•−^) can be generated from the mitochondrial electron transport chain, which is the main source of O_2_^•−^ and many enzymes, including nicotinamide adenine dinucleotide phosphate (NADPH) oxidase, xanthine oxidase, lipoxygenase, cyclooxygenase and cytochrome P450 reductase [9]. Once formed, O_2_^•−^ is involved in several reactions, which, in turn, may generate other ROS/RNS such as hydrogen peroxide (H_2_O_2_), hydroxyl radical (^•^OH), peroxynitrite (ONOO^−^) and hypochlorous acid (HOCl). Of note, O_2_^•−^ generated in the mitochondria is not able to cross the inner mitochondrial membrane, thus it is confined to the matrix where it reacts rapidly with the enzyme manganese superoxide dismutase (MnSOD) to form H_2_O_2_. H_2_O_2_, a non-radical ROS, is the only freely diffusible form of ROS able to penetrate membranes, and is also produced by other oxidases, including amino acid oxidase and xanthine oxidase. Thus, on the basis of subcellular compartmentation, mitochondrial ROS (mROS) accumulation is predicted to elicit different signals from H_2_O_2_ stimulation. ^•^OH, the most reactive among all free radical species, is generated by the reaction of O_2_^•−^ with H_2_O_2_ in the presence of Fe^2+^ or Cu^2+^ in the Fenton reaction [10]. Regarding non-enzymatic reactions, free radical production may occur when oxygen reacts with organic compounds or when cells are exposed to ionizing radiation. Moreover, as stated above, non-enzymatic free radical production can occur also during mitochondrial respiration [7].

RNS refer to a number of nitrogenous products such as nitric oxide (NO), nitroxyl (HNO), nitrosonium cation (NO^+^), higher oxides of nitrogen, S-nitrosothiols (RSNOs), ONOO^−^ and dinitrosyl iron complexes. NO is synthesized through arginine-to-citrulline oxidation by NO synthases (NOS) [11]. Three NOS isoforms have been identified and are named according to the cell type or the condition under which they were first described: endothelial NOS (eNOS), neuronal NOS (nNOS) and inducible NOS (iNOS) [11]. NO is generated by iNOS under many pathological conditions, and can rapidly react with O_2_^•−^ to generate the more toxic ONOO^−^ and the highly reactive ^•^OH, which may then react with DNA, RNA, proteins and membrane lipids. ONOO^−^ is a key factor in protein tyrosine nitration of important structural and functional domains, causing changes in protein functions [12]. 

The generation of ROS from molecular oxygen is a natural and indispensable part of aerobic life. Indeed, basal levels of ROS are essential for the exploitation of various cellular functions via the modulation of signal transduction pathways that regulate gene expression, support cell proliferation or induce cell death [13]. However, excessive levels of ROS induce cellular dysfunction and cause damage to cellular macromolecules such as DNA, lipids and proteins, eventually leading to cell death. Regarding ROS-dependent DNA damage, only ROS species produced within the nucleus or those capable of entering the nucleus, such as H_2_O_2_, can lead to DNA damage. In contrast, high O_2_^•−^ levels in the mitochondria or lysosomes do not. Along this line, many studies have demonstrated that increased ROS/RNS production supports the progression of inflammatory diseases [14,15,16] and promotes carcinogenesis as well as cancer progression [6,17,18]. Additionally, oxidative stress-mediated chronic inflammation is a key risk factor for tumorigenesis [19,20]. 

At the cellular level, ROS/RNS production is supported by the onset of an antioxidant detoxification response that is central to controlling and/or adapting to increased ROS/RNS levels [21,22]. The endogenous antioxidant detoxification system is a very complex response that relies on: (i) enzymatic antioxidants such as copper-zinc superoxide dismutase (CuZnSOD), manganese superoxide dismutase (MnSOD), glutathione peroxidases (GPxs) and catalase. CuZnSOD is widely distributed in the cytosol and nucleus, while MnSOD is present in the mitochondrial matrix. Both are able to efficiently scavenge for O_2_^•−^ and generate H_2_O_2_ [23,24,25], which, in turn, can be neutralized by GPxs and peroxiredoxins (thioredoxin-dependent peroxidases) [26,27]. (ii) Non-enzymatic antioxidants, such as vitamin E, vitamin C, carotenoids, flavonoids, selenium, thiol antioxidants (thioredoxin, lipoic acid and glutathione). (iii) Multiple regulatory factors, including the transcription factor nuclear factor erythroid 2-related factor 2 (NRF2) that induces gene expression by binding to the antioxidant responsive elements (ARE), nuclear factor-kB (NF-kB) and activator protein-1 (AP-1), which collectively regulate the expression of genes involved in the detoxification response [21,22]. The abnormal activity of the above enzymatic and non-enzymatic antioxidants is associated with the development of various human diseases, including cardiovascular, neurodegenerative and metabolic disorders as well as cancer [6,14,26,28,29,30].

## 3. The MEK5/ERK5 Signalling Pathway

The MAPK ERK5 is encoded by the MAPK7 gene [1,2] and is involved in survival, anti-apoptotic signalling, proliferation and differentiation of several types of cells [3]. In response to extracellular stimuli, such as growth factors [31,32,33,34] and cellular stresses [35,36,37], the Ser/Thr kinases MEKK2 and MEKK3 activate MEK5, a dual-specificity protein kinase that has ERK5 as its only known substrate. Once activated, MEK5 phosphorylates ERK5 at T219 and Y221 in the conserved threonine-glutamic acid-tyrosine (TEY) motif of the catalytic domain [38]. MEK5-dependent phosphorylation stimulates ERK5 nuclear translocation, which has been proposed as a key event in the regulation of cell proliferation [39,40,41]. Despite sharing high homology in the kinase domain with ERK2 and possessing a TEY motif identical to that of ERK1/2/8 in the activation loop, ERK5 has a long C-terminal tail that is unique among all MAPKs. The C-terminal tail includes a nuclear localization sequence (NLS) important for ERK5 nuclear shuttling, two proline-rich (PR) domains (PR1 and PR2) that are considered potential binding sites for src-homology 3 (SH3)-domain-containing proteins, a nuclear export sequence (NES) and a myocyte enhancer factor 2 (MEF2)-interacting region [42]. The C-terminus of ERK5 also possesses a transcriptional transactivation domain (TAD) [43] that undergoes autophosphorylation of serine and threonine residues, thereby enabling ERK5 to directly regulate gene transcription [44]. A number of C-terminal residues are also known to be phosphorylated by cyclin-dependent kinase (CDK) 1 and/or ERK1/2 [45,46,47,48,49]. Among the known substrates of ERK5 are the transcription factors c-FOS, c-MYC, SRF accessory protein-1a (Sap-1a), myocyte enhancer factor 2A (MEF2A), C and D, as well as other kinases such as ribosomal S6 kinase (RSK) and serum/glucocorticoid-regulated kinase (SGK) [50]. Several reports support the view that the kinase and transactivation activities of ERK5 can be uncoupled under certain circumstances. For example, it has been shown that the nuclear localization of an ERK5 mutant devoid of kinase activity results in the activation of transcription through the TAD located at the C-terminus [51]. On the other hand, some ERK5 inhibitors such as XMD17-109 and AX15836 have been reported to drive nuclear translocation of ERK5, leading to paradoxical activation of ERK5 transcriptional activity, despite effectively inhibiting its kinase activity [52]. On the other hand, some ERK5 inhibitors (e.g., AX15836 and BAY-885) do not show anti-proliferative or anti-inflammatory effects [53,54]. This fact may be due to the above-mentioned paradoxical ERK5 activation, or to the possibility that kinase-independent activities of ERK5 may play crucial roles in ERK5-elicited signals. However, several reports have clearly demonstrated inhibitory effects on cell proliferation upon ERK5 knockdown or knockout [3,4]. At variance with the above reports, Cook’s group observed that siRNA-mediated ERK5 knockdown had no effect on the proliferation of colon rectal cancer cell lines with either KRAS or BRAF mutations [55]. Moreover, it has been recently reported that INY-06-061, a potent and highly selective heterobifunctional degrader of ERK5, did not induce anti-proliferative effects in multiple cancer cell lines nor suppress inflammatory responses in primary endothelial cells (ECs) [56].

## 4. Activation of the MEK5/ERK5 Pathway by Oxidative Stress

The first evidence that ERK5 is a redox-sensitive kinase came from Abe and colleagues, who showed that H_2_O_2_ is a potent stimulus of ERK5 activation, as determined using a kinase assay (Figure 1). ERK5 activation following exposure to H_2_O_2_ turned out to be calcium-dependent, because depletion of intracellular calcium using thapsigargin prevented H_2_O_2_-induced ERK5 activation. These effects were observed in different cell types, including human umbilical vein endothelial cells (HUVECs), arterial smooth muscle cells and skin fibroblasts [35]. In a later study, the same authors demonstrated that SRC supports H_2_O_2_-induced ERK5 activation in mouse fibroblasts. Indeed, H_2_O_2_ failed to stimulate ERK5 activity in the presence of the src family kinase inhibitors (SFKi) herbimycin A and PP1, as well as in src^−/−^ mouse embryonic fibroblasts [57]. The involvement of SFK in H_2_O_2_-induced ERK5 activation has also been reported in rat aortic smooth muscle cells, in which src inhibition using either SFKi PP2 or src specific siRNA attenuated H_2_O_2_-induced ERK5 activation [58]. We can include the lack of SFK inhibition by phosphatases such as the low molecular weight protein phosphatase, which is inactivated by H_2_O_2_, among the upstream mechanisms that may be responsible for SFK activation by H_2_O_2_ [59]. Additional SRC-dependent mechanisms may also be involved in ERK5 activation. Indeed, among the known upstream activators of MEK5/ERK5 signalling, a number of oncogenic signalling molecules and pathways are activated by SRC, including Abelson murine leukaemia viral oncogene homolog 1 (ABL1), PI3K/AKT, RAS/RAF/MEK/ERK and STAT3 [60,61]. A study conducted using pheochromocytoma 12 (PC12) cells later reported that treatment with the SFKi herbimycin A or PP2 or overexpression of a kinase-inactive SRC mutant (K297R) prevented H_2_O_2_-induced ERK5 activation, further supporting the involvement of SFK in this event. While studying upstream ERK5 activators, Sun and colleagues found that H_2_O_2_ caused significant activation of MEKK2, and that MEKK2’s interaction with the adaptor protein LAD is necessary for H_2_O_2_-induced ERK5 activation in CCL64 mink epithelial cells. Pre-treatment of the same cells with the SFKi PP1 prevented MEKK2 and ERK5 activation, indicating that H_2_O_2_ stimulates an SFK/LAD/MEKK2/MEK5/ERK5 cascade [62]. A recent study confirmed the involvement of MEKK2 in H_2_O_2_-induced ERK5 activation as H_2_O_2_-induced phosphorylation/activation of ERK5 was impaired in MEKK2^−/−^ intestinal stromal cells [63]. 

Regarding additional mechanisms involved in ERK5 pathway activation by oxidative stress, it has been reported that exposure to H_2_O_2_ increases PI3K/AKT [64] and c-ABL activation [65,66]. On the other hand, H_2_O_2_ was reported to modulate specific cysteine residues within redox-sensitive proteins such as EGFR [67,68] and to stimulate the induction of VEGF, which, in turn, binds to and activates VEGFR [69]. Both PP1 and PP2A Ser/Thr phosphatases are involved in H_2_O_2_-induced ERK5 activation, since PP1/PP2Ai okadaic acid and calyculin A prevent this activation in HeLa and PC12 cells [70]. Besides the activation of ERK5 kinase activity, an additional mechanism that may be responsible for the activation of the MEK5/ERK5 pathway relies on the increase in ERK5 mRNA levels upon oxidative phosphorylation (OXPHOS) following pyruvate dehydrogenase kinase (PDK) inhibition by dichloroacetate (DCA) [71]. Finally, a recent study showed that laminar shear stress (LSS)-induced mtROS production leads to activation of the MEK5/ERK5/KLF2 pathway in ECs [72]. Interestingly, ionizing radiation, which is known to generate ROS, has been shown to cause sustained ERK5 activation and MEK5/ERK5 knockdown combined with irradiation markedly sensitizes one to radiotherapy by inducing strong inhibition of tumour growth in mouse xenografts [73,74].

**Figure 1 cells-12-01154-f001:**
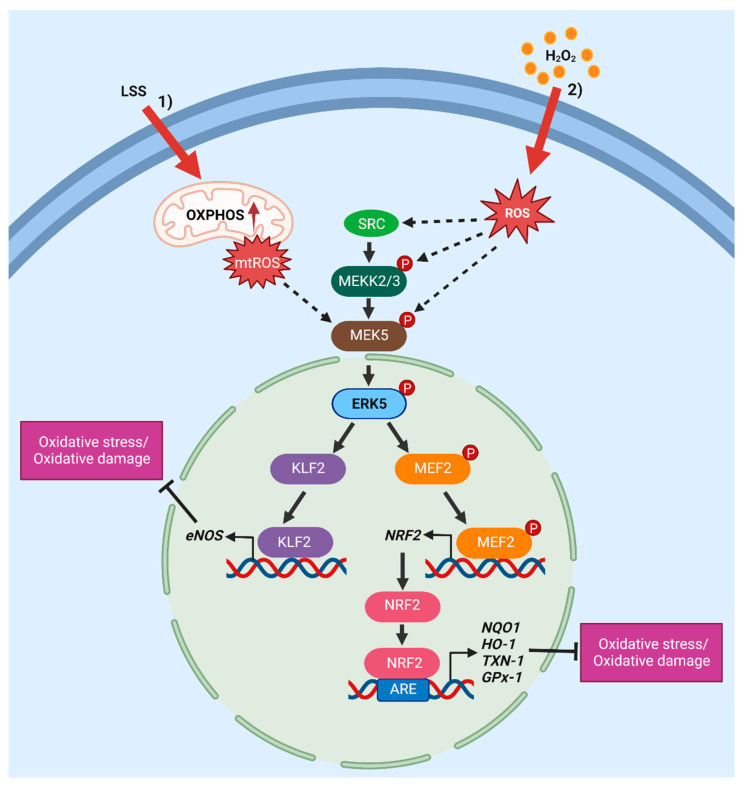
Overview of the mechanisms of activation and downstream effectors of the MEK5/ERK5 pathway in response to oxidative stress. The rising in oxidative stress triggers MEK5/ERK5 activation, which, in turn, promotes an antioxidant response. (1) Laminar shear stress (LSS) induces mtROS production, which, in turn, leads to activation of the MEK5/ERK5/KLF2 pathway [72]. (2) H_2_O_2_ stimulation may activate the ERK5 pathway via SRC [36,57,58,62], MEKK2/3 [63] or MEK5 [75,76] activation. Continuous arrows indicate direct activation mechanisms and dashed arrows indicate demonstrated, but not direct, mechanisms. eNOS, endothelial nitric oxide synthase; NQO1, NAD(P)H quinone dehydrogenase 1; HO-1, heme oxygenase-1; TXN-1, thioredoxin-1; GPx-1, glutathione peroxidase (created using Biorender.com accessed on 23 March 2023).

## 5. Role of the MEK5/ERK5 Pathway in ROS-Dependent Effects on Human Diseases

### 5.1. Pathophysiology of Blood Vessels

#### 5.1.1. Endothelial Cells

ECs play a critical role in cardiovascular homeostasis by regulating vascular tone and blood fluidity, fibrinolysis, angiogenesis as well as leukocyte and platelet adhesion [77]. NO has an important role in the regulation of vascular tone [78,79] and its deficiency, as much as excessive ROS (mainly O_2_^•−^), may promote endothelial dysfunction [80,81,82]. EC damage is associated with aging and the development of various cardiovascular disorders, including hypertension, atherosclerosis, heart failure and diabetes [83,84,85].

Erk5 knockout mice display cardiovascular defects, in particular those affecting heart development, vessel maturation, angiogenesis and endothelial integrity, which collectively lead to embryonic lethality at around stage E10 [86,87,88,89]. Analogous effects have been observed in mice in which Erk5 was selectively knocked out in ECs [87], thus confirming the critical role of ERK5 in normal cardiovascular development and vascular integrity. In this regard, in vitro studies have revealed that ERK5 is required to prevent apoptosis and mediate shear stress signalling in normal ECs, as well as to regulate tumour angiogenesis [90]. However, small molecule inhibitors of ERK5 do not seem to alter the vasculature in adult mice [91], leaving open the possibility of targeting ERK5 in human diseases.

The fact that ERK5 is activated by oxidative stress (i.e., H_2_O_2_) in ECs has been reported using HUVECs [39,75]. A later study provided evidence that ERK5 activation plays a protective role when ECs experience oxidative stress. Indeed, activation of ERK5 by overexpression of a constitutively active MEK5 mutant (MEK5-DD, S313D/T317D) protected HUVECs from H_2_O_2_-induced cell death. Depletion of NRF2 by specific siRNA abrogated these effects, pointing to the existence of an ERK5–NRF2 pathway. The same study also demonstrated that laminar flow induces ERK5–NRF2 pathway activation in HUVECs. Indeed, blocking MEK5/ERK5 using either siRNA (specific for ERK5) or BIX02189 (MEK5i [92]) inhibited laminar flow-induced upregulation of NRF2-dependent genes such as heme oxygenase-1 (HO-1) and NAD(P)H quinone dehydrogenase 1 (NQO1). Additionally, MEK5-DD-induced activation of ERK5 increased the transcriptional activity and nuclear translocation of NRF2, whereas BIX02189 strongly suppressed the latter. Immunoprecipitation analysis showed that ERK5 binds to NRF2 and that this interaction is diminished by a mutant form of ERK5 (ERK5-AEF, T218A/Y220F) that cannot be phosphorylated by MEK5 [76]. Interestingly, using the osteoblastic MC3T3-E1 cell line, Xia and colleagues demonstrated the involvement of ERK5/NRF2 signalling in the protective effect of mangiferin, a naturally available polyphenol found in both mango and papaya [93], in response to ROS. Indeed, genetic inhibition of ERK5 using specific siRNA abolished the anti-apoptotic effect of mangiferin upon H_2_O_2_ treatment, pointing to the involvement of ERK5 in the protective effect of mangiferin [94]. Another study reported that sustained activation of ERK5 following the overexpression of MEK5-DD suppresses the generation of ROS/RNS caused by growth factor deprivation in HUVECs. The same study also demonstrated that shear stress-activated ERK5 signalling restores the redox state of ECs by reducing ROS production and increasing NO bioavailability [95]. Additional evidence of the protective role of ERK5 in HUVECs was provided by showing that two natural antioxidants, the stilbenes resveratrol and pterostilbene, activate MnSOD expression through the ERK5/HDAC5 pathway, thus alleviating mitochondrial oxidative stress in ECs that may be eventually associated with cardiovascular disease [96].

Overall, the above studies established that ERK5 plays a protective role against oxidative stress-induced damage of ECs.

#### 5.1.2. Atherosclerosis

Atherosclerosis is a chronic inflammatory disease that is associated, in its early phase, with dyslipidaemia, increased expression of inflammatory factors and endothelial dysfunction, which collectively drive atherosclerotic lesion development in arteries [97,98]. 

Sugars have antiproliferative and proatherogenic effects on ECs and increase the risk of cardiovascular disease [99]. In order to better understand the mechanisms underlying hyperglycaemia-induced proatherogenic changes in ECs, Liu and colleagues found that glucose, and to a lesser extent raffinose, increase ERK5 activity in bovine pulmonary artery ECs. The involvement of ROS in glucose-induced ERK5 activation was proven by showing that N-acetylcysteine (NAC), a free radical scavenger, prevents the increase in ERK5 activation induced by glucose [100]. Another study conducted using ECs reported that despite activating ERK5 kinase activity, both H_2_O_2_ and advanced glycation end products (AGE) inhibited LSS-induced ERK5/MEF2/KLF2 and the subsequent eNOS expression through ERK5 SUMOylation. More importantly, ERK5 SUMOylation was increased in the aortas of diabetic mice in vivo. On that basis, the authors proposed that inhibiting ERK5 SUMOylation might be a new therapy for diabetes-mediated endothelial dysfunction and inflammation [101]. By contrast, Coon and colleagues found that LSS-induced mtROS production leads to activation, rather than inhibition, of the MEK5/ERK5/KLF2 pathway in ECs, pointing to the involvement of ERK5 in the anti-inflammatory response to LSS-dependent mtROS in ECs [72]. 

In another study conducted using human aortic ECs (HAECs), Wu and colleagues found that tumour necrosis factor-α (TNF-α) [102] plays a causal role in atherosclerosis by increasing intracellular ROS through the small GTPase Rac-1, which, in turn, stimulates the expression of VCAM-1 and ICAM-1—which play a major role in the initiation of early atherosclerosis—in HAECs via NF-kB. All these effects were reverted by treatment with atorvastatin, which was found to activate ERK5 in these cells in line with the well-known role of ERK5 in vasoprotection and the maintenance of endothelial integrity [86,87,88,89,103]. Moreover, inhibiting ERK5 using ERK5i XMD8-92 [91] or specific siRNA ablated the effects of atorvastatin and increased ROS production [104], further supporting the protective role of ERK5. Another study reported that TNF-α increases oxidative stress in HUVECs through generation of 4-hydroxy nonenal, a lipid peroxidation end product, and NADPH oxidase 4 (NOX4), which is involved in ROS production. TNF-α increased the levels of VCAM-1 and E-selectin, which, when overproduced, fuel the atherogenic process. Additionally, TNF-α reduced ERK5 phosphorylation and krüppel-like factor 2 (KLF2) mRNA and protein levels. Interestingly, butyrate, a short-chain fatty acid that has been shown to exert antioxidant, anti-inflammatory and other protective effects in atherosclerosis [105], prevented the proatherogenic effects of TNF-α (i.e., the increase in VCAM-1, E-selectin and oxidative stress) by reducing the levels of ROS—measured by quantifying 4-Hydroxynonenal (4-HNE)—and rescuing the activation of the ERK5/KLF2 pathway [106].

Lipopolysaccharides (LPSs) can stimulate immune cell activation and initiate inflammatory responses, contributing to endothelial dysfunction, an early event in atherosclerosis. Zhong and colleagues demonstrated that the antimalarial drug halofuginone suppressed LPS-induced oxidative stress in HUVECs, while stimulating the expression of the key atheroprotective transcription factor KLF2, a positive regulator of the antioxidant transcription factor NRF2 [107]. The latter effects were mediated by ERK5 because blocking ERK5 with XMD8-92 abolished the effect of halofuginone on KLF2 mRNA and protein [108]. Sun and colleagues showed that oscillatory shear stress (OSS), which is known to support the production of proinflammatory mediators in HUVECs, stimulates the production of mtROS (O_2_^•−^) and reduces the level of the lactate receptor GPR81, ERK5 phosphorylation and KLF2 mRNA and protein. Activation of GPR81 using physiological doses of lactate reduced oxidative stress by inhibiting the production of ROS and rescued OSS-induced reduced expression of KLF2, which was mediated by ERK5. Indeed, XMD8-92 abolished the rescue of KLF2 induced by lactate-mediated GPR81 activation [109].

Several lines of evidence indicate that hyperglycaemia enhances the proliferation of vascular smooth muscle cells (VSMCs), a critical step in the pathogenesis of atherosclerosis [110,111,112]. In this respect, Yu and colleagues found that intermittent high glucose enhances the proliferation and migration of primary rat VSMCs. The antioxidant rutin, a flavonoid found in many plants, inhibited the proliferation and migration of VSMCs by suppressing the phosphorylation of MEK1/2, ERK1/2, PI3K, NF-κB and ERK5, and the production of ROS under fluctuating glucose levels [113]. A later study identified the link between ERK5 and hyperglycaemia-related effects. Indeed, fluvastatin, the first fully synthetic statin that is effective at reducing total and low-density lipoprotein cholesterol [114], increased the mRNA and protein levels of NRF2 and NQO1 in VSMCs. This effect was dependent on MEK5/ERK5 because it was reduced upon MEK5 (BIX02189) or ERK5 (siRNA) inhibition. Treatment of VSMCs with fluvastatin suppressed AGE-dependent proliferation and this suppression was prevented by pre-treatment with BIX02189. Mechanistically, fluvastatin suppressed cyclin D1 and CDK4, and these effects were prevented by BIX02189 pre-treatment. In addition, overexpression of MEK5-DD diminished AGE-induced proliferation, while NRF2 depletion using specific siRNA restored AGE-induced proliferation. This study, therefore, showed that fluvastatin decreases AGE-dependent proliferation via the ERK5–NRF2 axis [115]. Another study conducted using primary rat VSMCs [116] reported that the ERK5 pathway may be involved in oxidant stress-stimulation of early growth response gene–1 (Egr-1), which has a key role in atherogenesis [117]. Indeed, insulin and oxidant stress, both as single agents and in combination, significantly increased the level of Egr-1 protein. In particular, inhibition of the ERK5 and ERK1/2 pathways with MEK1/2/5i UO126—but not inhibition of the ERK1/2 pathway alone with MEK1/2i PD98059—completely blocked the effect of oxidative stress on Egr-1 protein levels, thus suggesting that the MEK5–ERK5 pathway may be involved in oxidant stress stimulation of Egr-1 [116]. 

Altogether, the above studies point to a protective role for ERK5 against atherosclerosis.

#### 5.1.3. Hypertension

Hypertension is a primary risk factor for cardiovascular diseases such as stroke, heart attack, heart failure and aneurysm, and increased oxidant stress is strongly implicated in the pathogenesis of hypertension. Angiotensin (Ang) II and endothelin-1 (ET-1) are two potent vasoconstrictors that mediate pleiotropic vascular actions through distinct receptors [118,119]. The activation of MAPK by Ang II and ET-1 in VSMCs plays an important role in the vascular changes associated with hypertension [120,121,122]. Touyz and colleagues showed that Ang II stimulated NADPH oxidase activity, and increased intracellular H_2_O_2_ production as well as p38MAPK and ERK5 phosphorylation in primary rat VSMCs in a dose-dependent manner [123]. These phosphorylation events were reduced by IGF-1Ri and EGFRi (AG1024 and AG1478, respectively), and were almost abolished by the NADPH oxidase inhibitor diphenyleneiodonium or by the intracellular scavenger, tiron. These findings suggested that Ang II activates p38MAP kinase and ERK5 via redox-dependent cascades that are regulated by IGF-1R and EGFR transactivation. In a subsequent study conducted using primary human VSMCs, the same authors reported that Ang II and ET-1 stimulate the phosphorylation of p38MAPK, JNK and ERK5, but not ERK1/2, via redox-sensitive processes. Ang II activated p38, JNK and ERK5 primarily through NAD(P)H oxidase-generated O_2_^•−^, while ET-1 stimulated these kinases via redox-sensitive processes that involve mtROS [124]. Besides VSMCs, the involvement of a p38/ERK5 axis in the oxidative stress response has been reported by Morimoto and colleagues, who demonstrated that a p38/ERK5/BCL6B pathway creates a positive feedback loop that drives self-renewal of spermatogonial stem cells via ROS amplification [125].

Collectively, the above studies suggest that ERK5-mediated signals may contribute to the pathogenesis of hypertension.

### 5.2. Pathophysiology of the Heart

In the heart, redox signalling is involved in physiological processes, including excitation-contraction coupling and cell differentiation, as well as, when unbalanced, pathological occurrences such as adverse cardiac remodelling, fibrosis and heart failure. NADPH oxidases, NO synthase and mitochondria are sources of ROS relevant to heart diseases, concurring during both myocardial and vascular dysfunction [126].

#### 5.2.1. Ischemic Heart Disease

Ischemia-reperfusion (I/R) injury, an event occurring invariably in tissues when blood flow deprivation is followed by its restoration, is caused by excessive ROS, which are produced both during ischemia and upon reperfusion. Based on this, remarkable efforts have been directed to clarify the mechanisms that induce excess ROS production during I/R [127,128]. Using freshly explanted guinea pig hearts, it has been shown that ischemia stimulates the activation of p90RSK, src and ERK5, while reperfusion stimulates activation of p90RSK and ERK1/2. Ex vivo pre-treatment of the hearts with SFKi PP2 showed that SFKs are upstream activators of ERK5 in the context of ischemia. On the contrary, pre-treatment of the hearts with the antioxidant N-2-mercaptopropionyl glycine identified a major role for H_2_O_2_ in the stimulation of ERK1/2 and p90RSK after I/R, but only a partial role for H_2_O_2_ in ischemia-induced src and ERK5 activation [129].

#### 5.2.2. Myocardial Disease

Mitochondrial dysfunction and metabolic cardiomyopathy are critical factors that lead to myocardial diseases. In the latter, unremitting metabolic stress leads to cardiac lipid overload, insulin resistance, energy deficiency, increased ROS production and apoptosis over time, finally leading to deterioration in contractility and heart failure [130]. Additionally, diabetic patients have a nearly threefold increased risk of developing heart failure compared with age-matched non-diabetics [131]. 

The first report to determine the role of ERK5 in cardiac dysfunction was by Yan and colleagues, who showed that ERK5 activation inhibits cardiac apoptosis and the subsequent dysfunction via inhibition of the feedback loop between phosphodiesterase 3A and inducible cAMP early repressor [132]. A later study using a cardiomyocyte-specific Erk5 knockout mouse model demonstrated, for the first time, the in vivo role of ERK5 in regulating hypertrophic growth and preventing heart failure. In the same paper, the authors identified the transcription factor MEF2 as a downstream effector of ERK5 in regulating the hypertrophic response [133]. 

A link between ERK5, ROS and cardiomyocyte health was first identified in a study showing that genetic inhibition of ERK5 using specific siRNA boosted hypothermia-induced apoptosis by increasing the levels of a series of ROS (O_2_^•−^, H_2_O_2_, ^•^OH), the pro-apoptotic protein Bim and intracellular calcium in these cells [134]. A later study investigating the possible role of ERK5 in myocardial disease secondary to metabolic disorders found that expression of ERK5, MEF2A and MEF2D decreased in the hearts of obese (ob/ob) and diabetic (db/db) mice following a high-fat diet (HFD), eventually leading to cardiomyopathy. To identify the mechanisms linking ERK5 deficiency to cardiomyopathy, the authors used Erk5-CKO (cardiac knockout) and control (Erk5-Flox) mice and found that O_2_^•−^ production was increased in HFD-Erk5-CKO hearts with respect to the controls. Interestingly, in the cardiac fibres of Erk5-CKO mice on HFD, OXPHOS capacity and mitochondria organization were reduced, while the density of lipid droplets was increased compared with HFD-treated Erk5-Flox mice. Finally, analysis of murine primary cardiomyocytes found that free fatty acid stimulation induced ERK5 degradation through calpain-1, a cysteine protease whose activation is dependent on Gp91phox—a component of the NADPH oxidase complex—and triggered by O_2_^•−^ [135]. These results seem to indicate that the reduction in ERK5 levels that accompanies the onset of myocardial disease may be a consequence of ROS-induced calpain-1 activation following a HFD. In another study, Yu and colleagues studied myocardial hypertrophy induced by transverse aortic constriction in vivo, and H_2_O_2_ administration to H9C2 cells (derived from rat heart tissue) in vitro. These treatments induced oxidative stress and inflammation, which were associated with increased expression of miRNA (miR)-143-3p, a non-coding RNA that has been reported to regulate ERK5 expression [136]. However, whether or not ERK5 is involved in myocardial hypertrophy has not been determined [137].

Cameron and colleagues reported, for the first time, that ERK5 is expressed in platelets and functions as a redox switch to promote maladaptive platelet signalling during myocardial infarction (MI), a condition with greatly elevated ROS [138]. Indeed, they found that H_2_O_2_ activated ERK5 in isolated human platelets and that ERK5 is activated in platelets isolated from mice following MI induced by permanent ligation of the left anterior descending coronary artery. More importantly, platelet-specific ERK5^−/−^ mice showed less platelet activation, reduced MI size and improved post-MI heart function. A later study provided evidence that the platelet scavenger receptor CD36 promotes thrombosis under atherogenic conditions by generating a redox-regulated signalling pathway requiring ERK5. Moreover, genetic deletion of ERK5 in megakaryocytes and platelets decreased platelet adhesion, activation and accumulation in vitro. Finally, in two different models of in vivo arterial thrombosis, ERK5 deficiency in platelets abrogated the enhanced thrombosis seen under hyperlipidaemic conditions [139].

The incidence of cardiovascular diseases (CVD) is higher in HIV+ patients subjected to chimeric antigen receptor T-cell (cART) therapy. Experiments carried out using primary macrophages, which are involved in the proinflammatory conditions that support CVD, showed that cART treatment increases mtROS, which, in turn, activate p90RSK, which is responsible for the subsequent phosphorylation of ERK5 at S496. This event reduced ERK5 transcriptional transactivation activity, resulting in reduced NRF2 activity on ARE, as measured by evaluating the expression of the NRF2-dependent antioxidant genes thioredoxin-1, HO-1 and GPx-1. Therefore, treatments directed at increasing ERK5–NRF2–ARE activity may offer promising approaches for overcoming the higher CVD incidence in cART patients [140]. In a more recent study, the same authors identified the possible mechanisms responsible for the increased incidence of CVD in cancer survivors with respect to the rest of the population. Using murine primary monocytes/macrophages, they found that doxorubicin and ionizing radiation, which is frequently used in the treatment of cancer, upregulates p90RSK phosphorylation, which, in turn, supports ERK5 phosphorylation at the inhibitory residue S496. In particular, these events increased the activity of poly (ADP-ribose) polymerase, a DNA damage response (DDR)-related molecule, and led to subsequent NAD+ depletion and an increase in mtROS production, ultimately resulting in mitochondrial dysfunction. These effects, including ERK5 phosphorylation, were reversed by the p90RSK inhibitor FMK-MEA [141] and by the overexpression of a dominant/negative p90RSK mutant (K94A/K447A) in primary macrophages. Importantly, doxorubicin- and ionizing radiation-induced mtROS production were inhibited in primary macrophages derived from mice in which wt ERK5 had been replaced with the ERK5-S496A mutant using CRISPR/Cas9 technique. This study, therefore, established the importance of ERK5 inhibitory phosphorylation at S496 and identified possible strategies for preventing CVD in cancer survivors [142].

Overall, the above studies established that ERK5 plays a protective role against oxidative stress-induced myocardial diseases.

### 5.3. Pathophysiology of the Lung

Accumulating evidence indicates that oxidative stress plays an important role in the pathogenesis of various lung disorders, including asthma, acute injury and pulmonary fibrosis, as well as cancer onset and progression [143,144]. The large surface area of the pulmonary tissue together with continuous exposure to high levels of O_2_ and endogenous and exogenous oxidants contribute to high ROS and RNS production. In fact, oxidants in this tissue may be generated endogenously through local metabolic reactions (e.g., from mitochondrial electron transport during respiration or the activation of phagocytes) or may be derived from exogenous sources such as air pollutants and cigarette smoke [145,146,147]. With regard to the former, using C10 murine lung epithelial cells, Scapoli and colleagues found that asbestos fibres, the phagocytosis of which is known to determine oxidant production [148], caused sustained ERK5 phosphorylation compared with treatment with EGF or H_2_O_2_, which were used as positive controls. Inhibition of MEK1/2/5 by UO126 or SFK by PP2—as well as inhibition by H_2_O_2_ or EGF—blocked asbestos-induced ERK5 activation in C10 cells. The involvement of src was confirmed in experiments showing that asbestos-induced ERK5 phosphorylation/activation was prevented by the overexpression of a dominant/negative src (K296R/Y528F) mutant. Additionally, pharmacological inhibition of src (using PP2) or overexpression of a dominant/negative MEK5 mutant (MEK5-AA, S311A/T315A [38]) significantly inhibited asbestos-induced cell proliferation, thus demonstrating that asbestos supports lung cell proliferation via an src/MEK5-dependent pathway. These results indicate that ERK5 inhibition may be a useful strategy against mineral fibre-induced carcinogenesis [149]. 

A possible protective role for ERK5 activation by oxidative stress in the lungs has been identified using A549 lung adenocarcinoma cells. Indeed, Wu and colleagues showed that ERK5 mediates antioxidant response to the oxidant-sensitive miR-200c, the expression of which is induced by H_2_O_2_ [150]. In these cells, overexpression of miR-200c reduced the expression of three antioxidant proteins—SOD2, HO-1 and sirtuin 1—with respect to control cells, under basal or H_2_O_2_-stimulated conditions. ERK5 KD using specific shRNA reduced the expression of sirtuin 1, in keeping with the notion that ERK5 is an upstream regulator of this gene [151]. This study therefore provided evidence that ERK5 participates in the modulation of redox homeostasis of normal and neoplastic lung cells [152]. Another study showed that upregulation of the MEK5/ERK5/NRF2 antioxidant pathway plays an indispensable role in regulating the antioxidant effect of the isoflavone biochanin A, alleviating lung injury induced by water-soluble components of urban particulate matter. [153]. 

The above studies seem to indicate that ERK5 plays a protective role against oxidative stress-induced lung injury.

### 5.4. Pathophysiology of the Lymphohematopoietic System

In recent years, ROS have been proven to participate in the regulation of the lymphohematopoietic system, which is responsible for the development of immune response and the production of blood cells. Indeed, low ROS levels are important for the maintenance of quiescence and multipotency of haematopoietic stem cells, whereas a higher level of ROS may support haematopoietic differentiation [154,155]. Furthermore, similar to what is well established in solid cancers, high ROS levels and oxidative stress play a key role in leukaemia onset and progression by stimulating genomic instability, cell survival and proliferation as well as drug resistance [154,156].

In Jurkat acute T-lymphoblastic leukaemia cells, ERK5 KD using specific shRNA impaired OXPHOS and reduced cell survival [157]. In further support of a pro-survival role for ERK5 in the above cellular model, a later study showed that Fas ligand (FasL)-induced apoptosis was attenuated by H_2_O_2_-induced ERK5 activation. The protective effect of H_2_O_2_ was abolished in Jurkat cells by the overexpression of MEK5-AA. On that basis, the authors proposed that inhibiting ERK5 signalling in conjunction with conventional chemotherapy, which is known to induce ROS production, might be more effective at maximizing leukemic cell death compared with chemotherapy alone [158]. Along this line, a study by Khan and colleagues shed light on the mechanisms through which ERK5 modulates the antioxidant response in leukemic cells. In human acute leukaemia cell lines (Jurkat, OCI-AML3, NB4 and MOLM-13), OXPHOS elicited using either DCA or an “OXPHOS medium” (e.g., glucose-free culture medium with glutamine and galactose) induced an increase in NQO-1, HO-1 and ERK5 mRNA levels, while decreasing the levels of Keap-1, a key sensor of oxidative stress that mediates ubiquitination and degradation of the NRF2 protein. Similar results were obtained upon DCA treatment of cells derived from AML patients, and in NOD/SCID-interleukin-2 receptor γ null (NSG) mice engrafted with primary human leukemic cells derived from patients with different haematological neoplasms (e.g., multiple myeloma, B-cell chronic lymphocytic leukaemia and T-cell lymphoma). The authors found that MEF2C, a downstream target of ERK5, binds to the promoter of miR-23a–27a–24-2 cluster and that miR-23a destabilizes the Keap-1 mRNA. These results suggest that, in leukemic cells undergoing OXPHOS, ERK5/MEF2/miR-23a signalling downregulates Keap-1, leading to activation of the NRF2/ARE pathway, which, in turn, generates an antioxidant response [159]. In a later study, the same authors demonstrated that in addition to increasing ERK5 mRNA, DCA-induced OXPHOS supports NRF2 expression in Jurkat, OCI-AML3 and NB4 cells. Similar results were obtained upon DCA treatment of cells derived from AML patients and NOD/SCID-NSG mice engrafted with AML cells. Indeed, ERK5 KD using ERK5-specific shRNA reduced the levels of DCA-induced NRF2 mRNA and protein, as well as expression of the NRF2-target genes NQO-1 and HO-1. Conversely, NRF2 mRNA levels increased following the expression of MEK5-DD. ERK5-dependent regulation of NRF2 upon DCA treatment was dependent on MEF2, because MEF2-targeting siRNA reduced DCA-induced NRF2 expression. Inhibition of mitochondrial complex I by metformin decreased ERK5, NRF2 and NQO-1 mRNA and protein levels. Conversely, complex I activity induced ERK5/MEF2-dependent NRF2 expression through fumarate accumulation, thus supporting the antioxidant response [71]. 

Collectively, the above studies indicate that ERK5 may reduce the burden of ROS that would result in the death of leukaemia cells, and ERK5-dependent antioxidant responses may prevent additional ROS-mediated mutagenesis that could support leukaemia progression.

### 5.5. Pathophysiology of the Kidney

The kidney is an organ characterized by a high metabolic rate, in particular oxidation reactions in the mitochondria, which make it vulnerable to damage caused by further oxidative stress. Oxidative stress is increased in patients with renal impairment as a result of enhanced oxidant activity and reduced antioxidant capacity, and inflammation further amplifies oxidant generation and impairs antioxidant systems, worsening patient conditions [160]. Moreover, consequences of oxidative stress in patients with chronic kidney disease include NO production, increased Ang II activity, hypertension, atherosclerosis and anaemia [161].

Glomerulonephritis (GN) encompasses a subset of renal diseases characterized by immuno-mediated damage to glomeruli. Acute GN forms can be either primary renal diseases or secondary to diabetes, hypertension, cardiovascular diseases, anaemia, metabolic acidosis and systemic immune-mediated diseases. Most acute GN are considered progressive disorders, which, without timely therapy, progress to chronic GN characterized by progressive glomerular damage and tubulo-interstitial fibrosis, leading to reduced glomerular filtration rates [162]. 

A link between oxidative stress-induced degenerative processes of the kidney and ERK5 was provided by a study showing that treatment with eplerenone (a selective mineralocorticoid receptor antagonist) or tempol (a cell membrane-permeable radical scavenger, 4-hydroxy-2,2,6,6-tetramethylpiperidine-N-oxyl) prevented glomerular changes (i.e., cell proliferation and mesangial matrix expansion) and hypertension following the administration of aldosterone/1% NaCl to male Sprague-Dawley rats. Aldosterone/1% NaCl-treated animals had enhanced levels of ERK1/2, JNK and ERK5 phosphorylation. Both eplerenone and tempol prevented this enhancement, pointing to an ROS-dependent mechanism of activation of all the above MAPKs. However, whether ERK5 activation was associated with the observed detrimental changes or was an attempt to protect the cells from the ROS-induced effects was not addressed [163].

As mentioned above, Ang II activity may be increased as a consequence of oxidative stress. Using rat mesangial cells, Ishizawa and colleagues found that PDGF, a mitogen involved in the pathogenesis of GN, induced H_2_O_2_ and src-dependent ERK5 phosphorylation. Indeed, SFKi (herbimycin A, PP2) abolished PDGF-induced ERK5 activation. Pre-treatment with olmesartan, an Ang II type 1 (AT1) receptor blocker that provides renoprotection and suppresses oxidative stress, prevented PDGF-induced src and ERK5 activation. However, whether or not ERK5 plays a protective role following H_2_O_2_ induction was not addressed in the study. In the same study, ERK5 activation was shown to be necessary for PDGF-induced migration in rat mesangial cells, because ERK5 siRNA suppressed PDGF-stimulated cell migration [164]. Interestingly, in human hepatic stellate cells, ERK5 inhibition increases, rather than reduces, PDGF-induced migration [31]. In another study, Urushihara and colleagues investigated whether ERK5 is involved in the pathogenesis of chronic GN. Starting from the fact that the level of O_2_^•−^ was significantly increased at the glomerular of nephritic rats (uninephrectomized rats treated with the nephritogenic anti-Thy-1 mAb) with respect to control rats, H_2_O_2_ induced ERK5 phosphorylation in rat mesangial cells in vitro in a dose-dependent manner. Genetic inhibition of ERK5 using specific siRNA decreased H_2_O_2_-induced cell viability and soluble collagen secretion in rat mesangial cells, pointing to the involvement of ERK5 in the pathogenesis of chronic GN [165].

High glucose is known to generate ROS—H_2_O_2_ in particular—in glomerular cells [166,167]. Along this line, Suzaki and colleagues found that high glucose levels activated ERK5 and ERK1/2, but not p38, in the glomeruli of Otsuka Long Evans Tokushima Fatty rats, which display diabetic nephropathy at 52 weeks of age. High glucose-induced ERK5 activation in cultured rat mesangial cells was reduced by inhibiting SFK (with herbimycin-A or PP2) or depleting PKC (with GF109203X- or prolonged PMA treatment). Taken together, these results indicate that high glucose induces SFK- and PKC-dependent ERK5 activation in mesangial cells, both in vivo and in vitro [168]. Another study showed that overexpression of ERK5 in mouse kidneys provides protection against renal I/R injury [169].

In conclusion, although several lines of evidence seem to indicate that ERK5 can support GN, further studies are needed to define the role of ERK5 in kidney pathophysiology.

### 5.6. Pathophysiology of the Liver

The role of oxidative stress has been extensively studied in the liver, a tissue at high risk of oxidative stress due to its high metabolic activity. Moreover, chronic liver diseases are nearly always characterized by increased oxidative stress, regardless of the aetiology [170].

As has been described for leukaemia cells (see above), activation of OXPHOS by DCA has been reported to induce an increase in NRF2 and ERK5, NQO-1 and HO-1 mRNA levels in two hepatocellular carcinoma (HCC) cell lines (Huh7 and HepG2) and in primary human hepatocytes. ERK5 KD (using siRNA) in primary human hepatocytes and HCC cells impaired DCA-induced expression of NRF2 and the NRF2-target genes NQO-1 and HO-1. ERK5-dependent regulation of NRF2 following DCA treatment was shown to depend on MEF2, because MEF2-specific siRNA reduced DCA-induced NRF2 expression. Finally, inhibition of mitochondrial complex I with metformin decreased the expression of ERK5, NRF2 and NQO-1 mRNA and proteins. The study showed that complex I activity induced ERK5 expression via fumarate accumulation, leading to MEF2/NRF2-mediated antioxidant response [71]. Different conclusions were reached in another study, where ERK5 mRNA levels decreased upon treatment with salidroside, a natural compound with ROS-scavenging activity that protects against CCl4-induced liver injury [171].

### 5.7. Pathophysiology of the Central Nervous System

The central nervous system (CNS) is one of the most metabolically active compartments in the body and, even at rest, consumes 20% of the body’s oxygen supply [172]. The presence of high concentrations of polyunsaturated fatty acids and the selectivity of the blood–brain barrier, which prevents the uptake of some antioxidants like vitamin E, make the CNS highly susceptible to oxidative stress [173]. More importantly, excessive ROS are generated after cerebral I/R, leading to severe damage to brain cells, including both neurons and glia [174]. 

The brain expresses the highest levels of ERK5 [89]. ERK5 expression is maximal during early embryonic development and declines as the brain matures [175]. Of note, ERK5 and ERK5-regulated MEF2 gene expression contribute to brain-derived neurotrophic factor-promoted survival of developing—but not mature—cortical and cerebellar neurons [175,176].

#### 5.7.1. Degenerative Diseases

Oxidative stress has been implicated in the progression of neurodegenerative diseases, including Alzheimer’s [177] and Parkinson’s [178,179] diseases, as well as amyotrophic lateral sclerosis [180]. The first evidence of a link between ERK5 and oxidative stress in the CNS was reported by Suzaki and colleagues, who showed that ERK5 is activated by H_2_O_2_ in a concentration-dependent manner in PC12 cells, an established model of neurosecretion and neuronal differentiation. They also found that SFKi (herbimycin A and PP2) treatment or overexpression of a kinase-inactive src mutant (K297R) inhibits H_2_O_2_-induced ERK5 activation in PC12 cells. In addition, H_2_O_2_ treatment increased the ability of the ERK5-downstream target, MEF2C, to bind DNA. These findings suggest that c-src-mediated ERK5 activation by H_2_O_2_ may counteract oxidative stress-induced damage, likely through the activation of MEF2C transcription factor [36]. Another study conducted using PC12 cells reported that nerve growth factor (NGF) stimulation or low H_2_O_2_ dose preconditioning (PC) protects PC12 cells and mouse primary hippocampal neurons against high H_2_O_2_ dose-induced cell death. Both NGF and PC induced ERK5 phosphorylation and increased expression of the KLF4 transcription factor. Two MEK5i, BIX02188 [92] and BIX02189, abolished the neuroprotective effect of NGF and PC. Consistently, activation of ERK5 upon overexpression of MEK5-DD led to increased survival of cells experiencing H_2_O_2_-induced insult. The induction of KLF4 by NGF or PC was blocked by ERK5-targeting siRNA, pointing to the involvement of ERK5 in this induction. Importantly, overexpression of MEK5-DD or KLF4 in H_2_O_2_-stressed cells upregulated the expression of anti-apoptotic proteins such as Bcl-2 and neuronal apoptosis inhibitory protein-2, and downregulated the expression of pro-apoptotic proteins such as Bax and Caspase 1 [181]. Based on the above findings, it can be concluded that the ERK5/KLF4 pathway functions as a common mechanism for NGF and PC neuroprotection.

Another interesting aspect of the neuroprotective role of ERK5 in the CNS is its involvement in the action of melatonin, which is an antioxidant and anti-apoptotic agent that plays a neuroprotective role in different CNS injuries [182,183]. Liu and colleagues showed that melatonin prevented ROS and malondialdehyde production, induced SOD increase and prevented apoptosis in murine neuroblastoma N2a cells under hypoxia (i.e., 10 ppm O_2_; 0.001% O_2_). Furthermore, they found that melatonin induced ERK1/2 and ERK5 phosphorylation by upregulating the expression of the zinc uptake transporter Zip1, thus exerting a protective effect against hypoxia-induced apoptosis in these cells [184].

A possible role of ERK5 in the pathogenesis of Parkinson’s disease has also been identified. Potdar and colleagues showed that piceid, a resveratrol precursor, protects human dopaminergic-like SH-SY5Y cells from dopamine-induced cell death via activation of ERK1/2 and ERK5, which results in the inhibition of apoptosis, the latter being caused by oxidative stress following exposure to high dopamine concentrations. Piceid pre-treatment reduced caspase-3/7 activity and increased the expression of the anti-apoptotic Bcl-2 protein. Piceid-mediated protection against the dopamine-induced effect was attenuated by inhibition of ERK1/2 and ERK5 pathways using UO126 (MEK1/2/5i) or XMD8-92 (ERK5i). Thus, the protective effect of piceid seemed to be mediated by ERK1/2/5 activation [185]. Cavanaugh and colleagues investigated the contributions of ERK5 and ERK1/2 to the survival of the MN9D murine midbrain dopaminergic neuronal cell line under basal conditions and in response to 6-hydroxydopamine (6-OHDA), a neurotoxic synthetic organic compound used to induce lesions in the nigrostriatal dopaminergic system (a model of Parkinson’s disease). The authors observed that MEK1/2/5i UO126 decreased the survival of MN9D cells. Overexpression of a dominant/negative form of either ERK5 (ERK5-AEF) or MEK1 (K97A), the upstream activator of ERK1/2, mimicked the effect of UO126 on MN9D cells in reducing cell survival. Overexpression of MEK5-DD and wt ERK5 increased cell survival but did not inhibit 6-OHDA-induced toxicity. In contrast, overexpression of a constitutively active MEK1 mutant (S218D/S222D) decreased cell survival and inhibited 6-OHDA-induced cell death. Finally, overexpression of a constitutively active MEF2C mutant (S444D) increased cell survival and inhibited 6-OHDA-induced cell death, suggesting that MEF2C is important for MN9D cells survival, and is likely the downstream target of ERK5 in these events. Taken together, these results suggested that activation of both ERK5 and ERK1/2 promote MN9D cell survival, and that ERK1/2 also protects MN9D cells against oxidative stress [186]. Overexposure to Mn causes irreversible movement disorders with signs and symptoms similar, but not identical, to idiopathic Parkinson’s disease [187,188]. Using MN9D cells, Ding and colleagues found that Mn exposure increased apoptosis, ROS production and MEK5 and ERK5 protein expression. The inhibition of MEK5/ERK5 signalling using MEK5i BIX02189 increased Mn-dependent cell death and ROS production with respect to Mn treatment alone [173]. Additionally, MEK5/ERK5 signalling was proven to regulate cell responses to ROS and to exert a protective effect against Mn-induced cytotoxicity, likely via upregulation of Bcl-2 and downregulation of Bax [189].

#### 5.7.2. Cerebrovascular Diseases

ROS produced upon cerebral I/R can have severe and irreversible detrimental effects on both neurons and glia [174]. In this regard, the first in vivo evidence of a link between ERK5 and oxidative stress in the CNS was provided using a four-vessel occlusion model of Sprague–Dawley rats. A study by Wang and colleagues showed that ERK5 was rapidly (from 10 min to 1 day, peaking at 30 min) activated (phosphorylation at TEY) upon reperfusion of the hippocampus following 15 min of ischemia. The involvement of ROS in this activation was demonstrated by the fact that NAC pre-treatment prevented I/R-induced ERK5 activation. Additionally, ERK5 activation upon reperfusion was suppressed by SFKi (PP2), nifedipine (an L-Type Voltage-Gated Calcium Channel (LVGCC) blocker) or dextromethorphan, an N-methyl-D-aspartate (NMDA) receptor antagonist. These results suggest that I/R-induced activation of ERK5 might be mediated by oxidative stress-activated SFK through NMDA receptor and LVGCC in the rat hippocampus [190].

Based on the above studies, it is clear that the MEK5–ERK5 pathway plays a protective role against oxidative stress-induced CNS damage.

## 6. Conclusions

Oxidative stress and free radicals are generally known for their detrimental effects on human health, as they contribute to the initiation and progression of several pathologies, ranging from inflammatory diseases to cancer. Regarding the role of ERK5 and the possible side effects occurring following its inhibition in case it is targeted therapeutically [5,41,191], accumulating evidence indicates that, overall, the MEK5/ERK5 cascade exerts a suppressive role on oxidative stress under both physiological and pathological processes (Table 1). Based on this, in inflammatory diseases, the inhibition of ERK5 or its downstream targets is very likely to result in a detrimental effect as a consequence of the impairment of the antioxidant response, with potential worsening of the severity of inflammation. Conversely, impairment of the antioxidant response following MEK5/ERK5 pathway inhibition may result in a desirable effect following chemotherapeutic treatments to boost oxidative stress-dependent cell death. Similarly, inhibition of ERK5 may ameliorate hypertension. Nevertheless, the above studies clearly indicate that the role of the MEK5–ERK5 pathway as a protective factor against oxidative-induced damage may result in complex outcomes that should be investigated further in view of MEK5/ERK5-targeted exploitation in clinics.

## Figures and Tables

**Table 1 cells-12-01154-t001:** Role of the ERK5 pathway in response to oxidative stress in pathological contexts.

Tissue/System	Pathological Condition	Biological Outcome	Role	References
Cardiovascular system	Myocardial disease	ERK5 KD boosts hypothermia-induced injury/apoptosis of rat cardiomyocytes in vitro.	Protective	[134]
		Cardiomyocyte-specific ERK5 KO in mice leads to dampened contractility.	Protective	[135]
		Platelet-specific ERK5 KO mice show enhanced hyperlipidaemia-induced thrombosis.	Detrimental	[139]
Central nervous system	Degenerative disease	H_2_O_2_-induced SRC/ERK5 activation counteracts oxidative stress-induced damage through MEF2C in PC12 cells.	Protective	[36]
		ERK5/KLF4 activation mediates PC- and NGF-induced neuroprotection against high H_2_O_2_ dose-induced cell death in PC12 cells and mouse primary hippocampal neurons in vitro.	Protective	[181]
		Piceid protects SH-SY5Y cells from dopamine-induced cell death via ERK5 activation.	Protective	[185]
		Pharmacological inhibition of MEK5 increases Mn-dependent cell death and ROS production in MN9D cells.	Protective	[189]
Urinary system	Glomerulonephritis	ERK5 activation enhances glomerular ECM accumulation in rats with GN.	Detrimental	[165]
		ERK5 overexpression in mouse kidneys protects against renal I/R injury in vivo.	Protective	[169]
Blood vessel	Endothelial dysfunction	ERK5/NRF2 activation protects HUVECs from H_2_O_2_-induced cell death	Protective	[76]
		ERK5 activation suppresses ROS/RNS generation caused by growth factor deprivation and shear stress in HUVECs.	Protective	[95]
	Atherosclerosis	H_2_O_2_ and AGE inhibits ERK5/MEF2/KLF2 via ERK5 SUMOylation in HUVECs.	Protective	[101]
		LSS-induced mtROS promotes anti-inflammatory response via MEK5/ERK5/KLF2 in mouse ECs in vitro.	Protective	[72]
		Statins inhibit TNF-α-induced ROS in human ECs via ERK5 activation in vitro.	Protective	[104]
		Halofuginone promotes anti-inflammatory response via ERK5/NRF2 in HUVECs.	Protective	[108]
		Lactate-induced GPR81 activation reduces OSS-induced oxidative stress via ERK5/KLF2 in HUVECs.	Protective	[109]

KD, knockdown; KO, knockout; cART, chimeric antigen receptor T-cell; mtROS, mitochondrial ROS; PC, low H_2_O_2_ dose preconditioning; NGF, nerve growth factor; Mn, manganese; ECM, extracellular matrix; GN, glomerulonephritis; I/R, ischemia-reperfusion; AGE, advanced glycation end products; LSS, laminar shear stress; OSS, oscillatory shear stress.

## Data Availability

Not applicable.

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
