# Peer review of "Pathophysiological Impact of the MEK5/ERK5 Pathway in Oxidative Stress"

_cells, 2023, doi:10.3390/cells12081154_

Round 1

Reviewer 1 Report

The review focuses on a very interesting aspect in the world of ERK5, where knowledge is quite extensive, but still many issues remain elusive. The review is easy to follow and the layout is well organised especially in point 5. The figures are clear and well designed and the English is correct. Overall it is an excellent review that should be published. 

 Some minor points that could be considered by the authors but not mandatory as the work is excellent overall. 

A) Avoid the extra word. For example in line 167---with regard to ...coul be Regarding to. In line 196 ERK5 Knockout in mice....ERK5 Knockout mice 

B) Line 30 states that ERK5 is the latest discovered member of the MAPK family. This is not quite correct.  For example ERK8 was discovered in 2002. In this sense, it would probably be more accurate to include "last discovered member of the canonical MAPK". 

C) ERK5 has been linked to ionising radiation, which in addition to damaging DNA is a potent inducer of oxidative stress.   There are several evidences linking ERK5 to the cellular response to ionising radiation (Broustas CG et al., 2020 ; Jiang W, et al., 2019) and also to KLf2 (Sadhukhan R et al., 2020). . It would be nice if authos could speculate on a possible link between ionising radiation induced oxidative stress and ERK5. However, I understand that this is not a major focus of this review, so this suggestion may not be considered.

Author Response

Reviewer #1 (Reviewer Comments to the Author):

The review focuses on a very interesting aspect in the world of ERK5, where knowledge is quite extensive, but still many issues remain elusive. The review is easy to follow and the layout is well organised especially in point 5. The figures are clear and well designed and the English is correct. Overall it is an excellent review that should be published.

Replay: We are grateful to the thank the Reviewer for the positive comments on the manuscript.

Some minor points that could be considered by the authors but not mandatory as the work is excellent overall.

  1. A) Avoid the extra word. For example in line 167---with regard to ...could be Regarding to. In line 196 ERK5 Knockout in mice....ERK5 Knockout mice.

Reply: We thank the Reviewer for this advice, and the text has been changed as suggested (Revised manuscript, lines 186 and 226).

  1. B) Line 30 states that ERK5 is the latest discovered member of the MAPK family. This is not quite correct. For example ERK8 was discovered in 2002. In this sense, it would probably be more accurate to include "last discovered member of the canonical MAPK".

Reply: The Reviewer is perfectly right. In the revised manuscript the text has been changed as suggested (we have replaced the sentence “ERK5 is the last discovered member of the mitogen-activated protein kinase (MAPK) family” with “ERK5 is the last discovered member of the canonical mitogen-activated protein kinase (MAPK)” (Revised manuscript, line 30).

  1. C) ERK5 has been linked to ionising radiation, which in addition to damaging DNA is a potent inducer of oxidative stress. There are several evidences linking ERK5 to the cellular response to ionising radiation (Broustas CG et al., 2020; Jiang W, et al., 2019) and also to KLf2 (Sadhukhan R et al., 2020) . It would be nice if authors could speculate on a possible link between ionising radiation induced oxidative stress and ERK5. However, I understand that this is not a major focus of this review, so this suggestion may not be considered.

Reply: We agree with the Reviewer that this information could be of interest for the readers, so that we have briefly included a possible link between ionising radiation-induced oxidative stress and ERK5, and cited the indicated references (Revised manuscript, lines 198-202).

Reviewer 2 Report

In this review, Tusa et al. cover the role of the MEK5/ERK5 mitogen-activated protein kinase (MAPK) pathway in oxidative stress responses. A role of ERK5 in H2O2-induced oxidative stress responses has been proposed early on by Jun-ichi Abe’s group, who initially observed increased phosphorylation and kinase activity of ERK5 upon H2O2 stimulation. Since then, the relation between ROS and ERK5 phosphorylation has been studied by numerous groups in various cell and tissue systems. Tusa et al. now for the first time cover this topic comprehensively by discussing the various findings in the pathophysiology of various tissues sensitive to oxidative stress including blood vessels, heart, and the central nervous system among others. The topic of the article is principally interesting and the authors deserve credit for taking the challenge of attempting to summarize the literature available on the effects of ROS on ERK5 activation. Unfortunately, in its present form the article has several shortcomings, which need to be addressed.

1) Citations: The article quotes many unnecessary, inappropriate, outdated or less relevant articles containing poor data or data purely based on multifunctional MEK inhibitors such as PD98059 or U0126. This is problematic throughout the article but particularly disturbing in the section “oxidative stress” where many outdated reviews, reviews published in questionable low-impact journals (references 16, 20) or even editorials shorter than one page are quoted (reference 6). Since the article’s main target audience is likely researchers working on ERK5, the authors should better concentrate on a few comprehensive reviews written by experts in the field (e.g. Sies and Jones, Nat Rev Mol Cell Biol. 2020 21(7):363-383 or Sies et al., Nat Rev Mol Cell Biol. 2022 Jul;23(7):499-515) to facilitate further reading. Moreover, unnecessary quotations should vigorously be removed throughout the article (e.g. reference 109, which describes the purification of rutin). Moreover, whenever possible, multiple redundant reviews should be replaced by original articles. E.g. in section 3 line 118 two different reviews and one original article of the authors themselves are quoted for alternative C-terminal phosphorylation of ERK5 while four original papers could be quoted.

2) In the section “oxidative stress” the different forms of ROS, their properties and origins should be better explained. For instance, H2O2 is the only freely diffusible form able to penetrate membranes. Other forms of ROS e.g. O2.- produced as byproduct of respiratory oxidation in the mitochondria cannot leave their compartment unless converted into H2O2 by SOD2. Hence, due to compartmentation mitochondrial ROS (mROS) accumulation is predicted to elicit different signals than H2O2 stimulation. I consider this information important both with respect to a known positive role of mROS (but not of H202) in certain inflammatory processes such as inflammasome activation and with respect to ROS-mediated ERK5 activation. For instance, activation of the MEK5/ERK5/KLF2 pathway by fluid shear stress in endothelial cells has recently been shown to require mROS production (Coon et al., J Cell Biol (2022) 221 (7): e202109144). By contrast, Abe’s group observed that H2O2 stimulation could enhance ERK5 phosphorylation but inhibited ERK5-mediated gene expression (e.g. of KLF2) by triggering ERK5 sumoylation, which was proposed to contribute to an enhanced susceptibility of type II Diabetes patients to cardiovascular disease (Woo et al., Circ. Res. 2008, 102, 538-545). In light of the above-mentioned Coon study and the debate raised by Abe’s group itself whether H2O2 triggers ERK5 activation or inactivation (as discussed in the extended discussion of the Woo paper itself), the authors may not fully ignore discussion of these controversial data. Because of the ongoing controversy raised by these studies, the authors should generally abstain from equalizing H2O2-mediated ERK5 phosphorylation with ROS-mediated ERK5 activation and precisely define which form of ROS was applied in which cells in the quoted studies.

3) Figures:
-In Fig. 1 the authors should only integrate connections that clearly have been demonstrated. Albeit they quote studies that have shown that H2O2 can induce PI3K/AKT, modulate RTKs such as EGFR and induce JNK/Abl activation, none of these studies demonstrated a link between their observations and H2O2-induced ERK5 phosphorylation. Hence, an involvement of AKT, EGFR and ABL in ROS-mediated ERK5 phosphorylation is pure speculation and should be omitted from the figure. The authors should further specify what they mean with “ROS” in the figure and back up their claims by mentioning the respective references in the legends. I also would suggest omitting MEF2C from the figure as the dependency of ERK5-mediated KLF2 expression on MEF2 factors is controversial (see e.g. Sunadome et al.,
Dev. Cell 2011, 20, 192–205.)
-For figure 2, the authors should provide a legend. In its current form, the figure is difficult to comprehend. Since, the role of ROS in cancer is not further discussed in this review they should also consider omitting cancer from the figure. Is “SNC” a typo? Do the authors mean “CNS”?

4) General:

The wealth of information in the review is overwhelming. Generally, there is too little visual information accompanying the text (only two figures), which complicates reading. Since the original data in the quoted studies of the section about VSMC are extremely poor and provide at best indirect evidence for a connection between ROS, ERK5 phosphorylation and VSMC functionality, I suggest omitting this part completely. The authors also should consider omitting or at least shortening other parts. For instance, the paragraph about hypertension similarly contains many less convincing and poorly conducted studies from low-impact journals. Overall, the review would benefit from focussing section 5 to the discussion of fewer tissues e.g. of blood vessels, heart, the lymphohematopoietic system, and the central nervous system as most studies dealing with the connection between ROS and ERK5 have been performed on these tissues. Since the authors force the idea of a suppressive role of the MEK5/ERK5 cascade on oxidative stress I suggest summarizing the most relevant studies from the different tissue systems that support this hypothesis in a table.

5) Minor:

-Page 2 line 105: the MEK5 phosphorylation sites in ERK5 are T219 and Y221, not T218 and Y220.
-Page 3, line 133: Since the effect of ERK5 on proliferation a matter of debate (see e.g. Lochhead et al., Cell Cycle. 2016; 15(4): 506–518 and You et al., Cell Chem Biol. 2022 Nov 17;29(11):1630-1638.e7) the authors should rephrase this sentence and include these citations.
- Page 3 line 102/103: In the context of growth factor-induced ERK5 activation at least Kato et al, Nature. 1998;395 (6703):713-6 should additionally be quoted. For stress-activated ERK5 activation the authors should include Yan et al., J Biol Chem. 1999; 274: 143–150, who first showed an effect of fluid shear stress on ERK5 activation.
- Page 5, line 198: As four different groups simultaneously have reported the phenotype of ERK5 knockout mice (for an overview see Hayashi et al. J Mol Med. 2004 Dec;82(12):800-8), all of them should be properly acknowledged. Reference 83 is not a study reporting Erk5 knockouts but describes the phenotype of Mek5 knockout mice.
- The review contains many unjustified speculations that are not based on facts. For instance, in line 614 the authors claim that “oxidative stress has a primary role in neurodegenerative disease” or line 600 “As a consequence of high oxygen consumption, a large amount of ROS/RNS may be generated in the brain and result in oxidative stress”. Such non-factual speculations should be removed (see also point 3, my criticism on Fig.1).

Author Response

Reviewer #2 (Reviewer Comments to the Author):

In this review, Tusa et al. cover the role of the MEK5/ERK5 mitogen-activated protein kinase (MAPK) pathway in oxidative stress responses. A role of ERK5 in H2O2-induced oxidative stress responses has been proposed early on by Jun-ichi Abe’s group, who initially observed increased phosphorylation and kinase activity of ERK5 upon H2O2 stimulation. Since then, the relation between ROS and ERK5 phosphorylation has been studied by numerous groups in various cell and tissue systems. Tusa et al. now for the first time cover this topic comprehensively by discussing the various findings in the pathophysiology of various tissues sensitive to oxidative stress including blood vessels, heart, and the central nervous system among others. The topic of the article is principally interesting and the authors deserve credit for taking the challenge of attempting to summarize the literature available on the effects of ROS on ERK5 activation. Unfortunately, in its present form the article has several shortcomings, which need to be addressed.

Replay: We would like to thank this Reviewer for his/her careful reading of the manuscript and the overall comment.

1) Citations: The article quotes many unnecessary, inappropriate, outdated or less relevant articles containing poor data or data purely based on multifunctional MEK inhibitors such as PD98059 or U0126. This is problematic throughout the article but particularly disturbing in the section “oxidative stress” where many outdated reviews, reviews published in questionable low-impact journals (references 16, 20) or even editorials shorter than one page are quoted (reference 6). Since the article’s main target audience is likely researchers working on ERK5, the authors should better concentrate on a few comprehensive reviews written by experts in the field (e.g. Sies and Jones, Nat Rev Mol Cell Biol. 2020 21(7):363-383 or Sies et al., Nat Rev Mol Cell Biol. 2022 Jul;23(7):499-515) to facilitate further reading. Moreover, unnecessary quotations should vigorously be removed throughout the article (e.g. reference 109, which describes the purification of rutin). Moreover, whenever possible, multiple redundant reviews should be replaced by original articles. E.g. in section 3 line 118 two different reviews and one original article of the authors themselves are quoted for alternative C-terminal phosphorylation of ERK5 while four original papers could be quoted.

Replay: We agree with the Reviewer that UO126 has been reported to inhibit MEK1/2/5; regarding PD98059, despite initially reported to similarly inhibit MEK1/2 and MEK5, it has been later demonstrated that it is more specific for MEK1/2 over MEK5 by Mody and colleagues and later confirmed by us (FEBS Lett. 2001, 502, 21-24; J. Immunol. 2008, 180, 4166-4172). Nevertheless, we agree that the possibility that the different specificity between UO126 and PD98059 allows the dissection of the involvement of MEK1/2 with respect to MEK5 remains questionable. This latter sentence is included in the text (Revised manuscript, lines 358-362). Moreover, we checked accurately the content of the papers that referred to the use of these MEKi, and removed the results obtained in the case they were not properly supported such as in the Suzaki study (Kidney Int. 2004, 65, 1749-1760) (lines 175-178, line 625, lines 678-679, and lines 682-684).

We thank the reviser for the suggestion regarding the quoted references. Based on the reviewer’s suggestion, we removed some references (Oxid Med Cell Longev. 2016, 2016, 5010423; Curr Pharm Des. 2018, 24, 4771-4778; Curr. Cancer Drug Targets 2018, 18, 538-557; Scientific World Journal 2013, 2013, 162750; Mol. Cell. Biochem. 2004, 266, 37-56; Int. J. Biochem. Cell Biol. 2007, 39, 44-84; Physiol. Rev. 2002, 82, 47-95; Crit. Rev. Food Sci. Nutr. 2004, 44, 275-95; Physiol. Rev. 2007, 87, 315-424; Cell Signal. 2007, 19, 1807-1819; Med. Sci. Monit. 2001, 7, 801-819; Moscow Univ. Biol. Sci. Bull. 2018, 73,199–202) (old references: 6-16,20), and replaced them with more relevant reviews written by experts in the field, including the two suggested by the Reviewer (Nat. Rev. Mol. Cell. Biol. 2020, 21,363-383; Nat. Rev. Mol. Cell Biol. 2022, 23, 499-515; Cancer Cell. 2020, 38, 167-197; Mol. Cell. 2021, 81, 3691-3707; Int. J. Mol. Sci. 2023, 24, 1841; Eur. Heart J. 2012, 33, 829-837; Nat. Rev. Drug Discov. 2021, 20, 689-709; Circ. Res. 2018, 122, 877-902). Regarding ERK5 C-terminal phosphorylation, we replaced then two reviews (Int. J. Mol. Sci. 2021, 22, 7594; Drug Discov Today. 2016, 21,1654-1663) with the original articles (PLoS ONE 2015, 10, e0117914; Cell. Death. Dis. 2016, 7, e2415; J. Cell. Sci. 2010, 123, 3146–3156; Cell. Signal. 2010, 22, 1829–1837) (line 130). Finally, the reference about the rutin (J. Agric. Food Chem. 1999, 47, 4649-4652) has been removed in the revised manuscript.

2) In the section “oxidative stress” the different forms of ROS, their properties and origins should be better explained. For instance, H2O2 is the only freely diffusible form able to penetrate membranes. Other forms of ROS e.g. O2.- produced as byproduct of respiratory oxidation in the mitochondria cannot leave their compartment unless converted into H2O2 by SOD2. Hence, due to compartmentation mitochondrial ROS (mROS) accumulation is predicted to elicit different signals than H2O2 stimulation. I consider this information important both with respect to a known positive role of mROS (but not of H202) in certain inflammatory processes such as inflammasome activation and with respect to ROS-mediated ERK5 activation. For instance, activation of the MEK5/ERK5/KLF2 pathway by fluid shear stress in endothelial cells has recently been shown to require mROS production (Coon et al., J Cell Biol (2022) 221 (7): e202109144). By contrast, Abe’s group observed that H2O2 stimulation could enhance ERK5 phosphorylation but inhibited ERK5-mediated gene expression (e.g. of KLF2) by triggering ERK5 sumoylation, which was proposed to contribute to an enhanced susceptibility of type II Diabetes patients to cardiovascular disease (Woo et al., Circ. Res. 2008, 102, 538-545). In light of the above-mentioned Coon study and the debate raised by Abe’s group itself whether H2O2 triggers ERK5 activation or inactivation (as discussed in the extended discussion of the Woo paper itself), the authors may not fully ignore discussion of these controversial data. Because of the ongoing controversy raised by these studies, the authors should generally abstain from equalizing H2O2-mediated ERK5 phosphorylation with ROS-mediated ERK5 activation and precisely define which form of ROS was applied in which cells in the quoted studies.

Replay: We thank the Reviewer for this comment that stimulated us to improve the text of the “Oxidative Stress” section (Revised manuscript: lines 51-54, lines 58-70). With regard to the positive role of mROS, but not of H202, in certain inflammatory processes and with regard to ROS-mediated ERK5 activation, the Coon study about the activation of the MEK5/ERK5/KLF2 pathway by fluid shear stress-induced mtROS production has been added in the revised manuscript as suggested by the Reviewer (Revised manuscript: lines 197-198, and lines 288-292). Finally, when possible, we have consistently replaced the world ROS with the specific form of ROS involved in the study (e.g. line 323, line 373, line 382, line 602, and line 606).

3) Figures:

-In Fig. 1 the authors should only integrate connections that clearly have been demonstrated. Albeit they quote studies that have shown that H2O2 can induce PI3K/AKT, modulate RTKs such as EGFR and induce JNK/Abl activation, none of these studies demonstrated a link between their observations and H2O2-induced ERK5 phosphorylation. Hence, an involvement of AKT, EGFR and ABL in ROS-mediated ERK5 phosphorylation is pure speculation and should be omitted from the figure. The authors should further specify what they mean with “ROS” in the figure and back up their claims by mentioning the respective references in the legends. I also would suggest omitting MEF2C from the figure as the dependency of ERK5-mediated KLF2 expression on MEF2 factors is controversial (see e.g. Sunadome et al., Dev. Cell 2011, 20, 192–205.)

Replay: Figure 1 has been modified on the basis of the above suggestions. In particular, i. We have removed AKT, EGFR and ABL among the signalling molecules possibly involved in ROS-mediated ERK5 phosphorylation; we agree, indeed, that these were mere speculations based on the inclusion of known upstream activators of ERK5 that have been demonstrated to be involved in ROS signalling; ii. we have specified the kind of ROS indicating the respective references in the figure legend (Revised manuscript: lines 205-209); iii we have omitted to depict the dependency of ERK5-mediated KLF2 expression on MEF2 factors.

-For figure 2, the authors should provide a legend. In its current form, the figure is difficult to comprehend. Since, the role of ROS in cancer is not further discussed in this review they should also consider omitting cancer from the figure. Is “SNC” a typo? Do the authors mean “CNS”?

Replay: We are really grateful to this Reviewer for the comment on Figure 2, that stimulated us to replace this figure with a more informative new Table (Table 1) where we summarized the most relevant studies about the role of the ERK5 pathway in the response to oxidative stress in different tissue systems, as suggested by the Reviewer. Similarly to the original aim of the former Figure 2, this new table indicates that the MEK5/ERK5 cascade exerts overall a suppressive role on oxidative stress in both physiological and pathological processes (Revised manuscript: lines 762-764).

4) General: The wealth of information in the review is overwhelming. Generally, there is too little visual information accompanying the text (only two figures), which complicates reading. Since the original data in the quoted studies of the section about VSMC are extremely poor and provide at best indirect evidence for a connection between ROS, ERK5 phosphorylation and VSMC functionality, I suggest omitting this part completely. The authors also should consider omitting or at least shortening other parts. For instance, the paragraph about hypertension similarly contains many less convincing and poorly conducted studies from low-impact journals. Overall, the review would benefit from focussing section 5 to the discussion of fewer tissues e.g. of blood vessels, heart, the lymphohematopoietic system, and the central nervous system as most studies dealing with the connection between ROS and ERK5 have been performed on these tissues. Since the authors force the idea of a suppressive role of the MEK5/ERK5 cascade on oxidative stress I suggest summarizing the most relevant studies from the different tissue systems that support this hypothesis in a table.

Replay:

We thank the Reviewer for the accurate reading of the text. As suggested by the Reviewer, we have summarized the most relevant studies about the role of the ERK5 pathway in the response to oxidative stress in different tissue systems in a new table (Table 1), in order to highlight the most significant findings in the field (see also the reply to the previous point). In line with the former Figure 2, this new table clearly indicates that the MEK5/ERK5 cascade exerts overall a suppressive role on oxidative stress in pathological processes. On the other hand, regarding the VSMC-related studies and those focused on  hypertension, since in the Cells journal there are no specific restriction in paper length nor in the journals in which the quoted literature should have been published, we felt that it is more correct to quote all the available literature irrespectively of the journal in which the studies have been published, provided they are Scopus/ISI journals indexed in PubMed.

5) Minor:

-Page 2 line 105: the MEK5 phosphorylation sites in ERK5 are T219 and Y221, not T218 and Y220.

Replay: We apologize for the oversight, and we have replaced the T218 and Y220 sites with T219 and Y221 sites in the revised manuscript (line 116).  

-Page 3, line 133: Since the effect of ERK5 on proliferation a matter of debate (see e.g. Lochhead et al., Cell Cycle. 2016; 15(4): 506–518 and You et al., Cell Chem Biol. 2022 Nov 17;29(11):1630-1638.e7) the authors should rephrase this sentence and include these citations.

Replay: With regard to the debate about the effect of ERK5 on proliferation, the citations suggested by the Reviewer have been added in the revised manuscript where appropriate (new text: lines 146-151).

- Page 3 line 102/103: In the context of growth factor-induced ERK5 activation at least Kato et al, Nature. 1998;395 (6703):713-6 should additionally be quoted. For stress-activated ERK5 activation the authors should include Yan et al., J Biol Chem. 1999; 274: 143–150, who first showed an effect of fluid shear stress on ERK5 activation.

Replay: We thank the Reviewer for this advice. Done as requested (lines: 113-114).

- Page 5, line 198: As four different groups simultaneously have reported the phenotype of ERK5 knockout mice (for an overview see Hayashi et al. J Mol Med. 2004 Dec;82(12):800-8), all of them should be properly acknowledged. Reference 83 is not a study reporting Erk5 knockouts but describes the phenotype of Mek5 knockout mice.

Replay: We have included in the revised manuscript the other two studies that have reported the phenotype of ERK5 knockout mice (Regan et al. Proc Natl Acad Sci U S A. 2002 Jul 9;99(14):9248-53; Yan et al. BMC Dev Biol. 2003 Dec 16;3:11) (line 228) and removed old reference 83 (Wang et al. Mol Cell Biol. 2005 Jan;25(1):336-45).

- The review contains many unjustified speculations that are not based on facts. For instance, in line 614 the authors claim that “oxidative stress has a primary role in neurodegenerative disease” or line 600 “As a consequence of high oxygen consumption, a large amount of ROS/RNS may be generated in the brain and result in oxidative stress”. Such non-factual speculations should be removed (see also point 3, my criticism on Fig.1).

Replay: In the revised manuscript, we have replaced the sentence “oxidative stress has a primary role in neurodegenerative disease” with “oxidative stress has been implicated in the progression of neurodegenerative diseases” (lines 671-672). Moreover, as suggested by the Reviewer, we have removed from the revised manuscript the sentence “As a consequence of high oxygen consumption, a large amount of ROS/RNS may be generated in the brain and result in oxidative stress”.

Round 2

Reviewer 2 Report

While Tusa et al. have addressed several of my concerns about their initial manuscript, there are still several shortcomings that require attention prior acceptance:

1)      Lines 37 onwards: Considering the controversial evidences that oxidative stress can both activate and inactivate EKR5 (e.g. by sumoylation),  I suggest changing “… by which oxidative stress promotes MEK5/ERK5 activation,…“ to “…by which oxidative modulates /influences MEK5/ERK5 activation,…“ and alter the following phrase “…stress-induced MEK5/ERK5 activation…“  to “…oxidative stress-dependent effects on MEK5/ERK5 activity…“.

2)      Section oxidative stress: Albeit is improved over the initial version and the authors now acknowledge a key role of compartmentation and different forms of ROS, it still contains too many general statements. For instance, statements such as „…excessive levels of ROS… cause damage to cellular macromolecules such as DNA…“ should be further specified as due to compartmentation only ROS species produced within the nucleus or those capable of entering the nucleus such as H2O2 can lead to DNA damage. By contrast, high O2.- levels produced in the mitochondria or in lysosomes do not. Some of the revised text further appears to be double edited (as indicated by different text colours) and is hard to comprehend  (line 49 onwards: ” Regarding O2.- it can be generated…”).  The authors should re-check and edit these passages for better readability.  In line 66 there is also a citation missing.

3)      Section 3,  line 117: Since there is still an ongoing debate whether ERK5 indeed is necessary for proliferation, I feel that writing “… ERK5 nuclear localization is a key event for the regulation of cell proliferation” is an overstatement, which should be omitted. Parts of this section, in particular the text from line 135 onwards should be revised to discuss the role of ERK5 in proliferation in a more unbiased manner. While I understand that based on their own data the authors favour the idea that ERK5 is essential for proliferation it is not Cook‘s group alone that claims that ERK5 is not essential for proliferation (see e.g. citation 56 or Lin et al., PNAS  (2016)  113(42):11865-11870), which should be adequately acknowledged.

4)      Section 4: The text from line 180 onwards is partially redundant to that in the previous section (from 165 onwards). The authors should combine both sections and discuss the potential role of ROS-sensitive RTKs such as EGF in the SRC-dependent regulation of ERK5 by H2O2 in a single section. There is nothing wrong with speculating that H202-mediated activation of EGF or VEGF may mediate the observed SRC-dependent ERK5 phosphorylation by H2O2 but such speculations should be clearly marked as such since formally this has not been shown. I further suggest citing the respective papers that have shown EGF- and VEGF-induced ERK5 activation in this context as the informed reader is probably aware that EGF and VEGF have previously been shown to activate ERK5 but an uninformed student reading the review is unlikely to make this connection.

5)      Fig.1: While figure 1 is improved from its initial version, the shown link from DCA to mtROS is at least misleading. In fact, the authors even have shown in their original paper that the observed ERK5 activation by mitochondrial damage is not due to enhanced mtROS (see e.g. Fig. 4 or 6 of the original article), which they also explicitly state in the discussion section of their paper.

6)      Line 226: The authors should omit “has been firstly reported” from the sentence. The publication that for the first time has shown H2O2-induced ERK5 phoshorylation in HUVEC  was much earlier  (citation 35).

7)      Line 272 and following: The authors should remove the statement that mtROS has an anti-inflammatory effect. In fact, mtROS production is a key event in NLRP3-dependent inflammasome activation (see Zhou et al., Nature (2011), 469(7329):221-5) and thus hardly can be called  “anti-inflammatory”.

8)      Line 330 and following: As criticized in the original review, the discussion of data based on the dual-specific inhibitors PD98059 and UO126 does not allow meaningful interpretations. I thus suggest omitting the whole paragraph from line 331 onwards.

9)      Section 5.2.2. As criticized in my initial review, the general use of terms such as “ROS” should be avoided throughout the article. In the section from line 400 onwards  (lines 402, 409, 415) again there is no further specification, which forms of ROS are induced by hypothermia or high fat diet. At least the readouts for ROS detection in these papers should be given to allow a judgement if mtROS or other forms of ROS are supposedly relevant.

10)   Lines 507 onwards: Again, the statement that “…high ROS levels (which ones?) and oxidative stress (which form?) play a key role in leukaemia onset and progression, by stimulating genomic instability…” is too general. The authors should rephrase this statement.
